# A Multi-Scale Object Detector Based on Coordinate and Global Information Aggregation for UAV Aerial Images

Liming Zhou [1,2], Zhehao Liu [1,2], Hang Zhao [1,2], Yan-E Hou [1,2,*], Yang Liu [1,2,3], Xianyu Zuo [1,2] and Lanxue Dang [1,2]

1   Henan Key Laboratory of Big Data Analysis and Processing, Henan University, Kaifeng 475004, China; lmzhou@henu.edu.cn (L.Z.); liuzhehao@henu.edu.cn (Z.L.); bless0929@henu.edu.cn (H.Z.); sea@vip.henu.edu.cn (Y.L.); xianyu_zuo@henu.edu.cn (X.Z.); danglx@vip.henu.edu.cn (L.D.)
2   School of Computer and Information Engineering, Henan University, Kaifeng 475004, China
3   Henan Province Engineering Research Center of Spatial Information Processing and Shenzhen Research Institute, Henan University, Kaifeng 475004, China
*   Correspondence: houyane@henu.edu.cn

**Abstract:** Unmanned aerial vehicle (UAV) image object detection has great application value in the military and civilian fields. However, the objects in the captured images from UAVs have problems of large-scale variation, complex backgrounds, and a large proportion of small objects. To resolve these problems, a multi-scale object detector based on coordinate and global information aggregation is proposed, named CGMDet. Firstly, a Coordinate and Global Information Aggregation Module (CGAM) is designed by aggregating local, coordinate, and global information, which can obtain features with richer context information. Secondly, a Feature Fusion Module (FFM) is proposed, which can better fuse features by learning the importance of different scale features and improve the representation ability of multi-scale features by reusing feature maps to help models better detect multi-scale objects. Moreover, more location information of low-level feature maps is integrated to improve the detection results of small targets. Furthermore, we modified the bounding box regression loss of the model to make the model more accurately regress the bounding box and faster convergence. Finally, we tested the CGMDet on VisDrone and UAVDT datasets. The proposed CGMDet improves mAP0.5 by 1.9% on the VisDrone dataset and 3.0% on the UAVDT dataset.

**Keywords:** UAV images; multi-feature fusion; information aggregation; multi-scale object detection

## 1. Introduction

UAV application technology has also made significant progress in recent years. Due to good mobility, convenient use, and low cost, UAVs have an extremely high application value in disaster monitoring [1], geological investigation [2], air traffic control [3], emergency relief [4], and other aspects. Therefore, UAV image object detection has been paid more attention by researchers. However, as the shooting angle and height of UAV are changeable, UAV object detection faces the two following challenges: (1) Due to the problem of UAV shooting perspective, there are considerable differences between the scales of targets of the same category or different categories, and there are many small objects, which greatly test the performance of the model to detect multi-scale targets and small targets. (2) In UAV images, there are usually many objects that are blocked, and weak light results in the poor visibility of boundary and features of the object, so it is hard to extract discriminant features from the model.

With the development of deep learning, traditional object detection methods, such as HOG [5] and SIFT [6], are gradually eliminated due to the need for a large amount of prior knowledge. However, the object detection method based on deep learning does not need the manual involvement and can dig deeper and more abstract features. Although

the object detector based on deep learning has achieved great results in detecting natural images, there are still great challenges for the object detection of UAV images.

To resolve these problems, this paper proposed a multi-scale object detector based on coordinate and global information aggregation, named CGMDet. Firstly, we designed the CGAM, which can make the model focus on coordinate and global information to alleviate the interference brought by the background. Secondly, we proposed the FFM to better fuse the features of different scales and add the feature maps of larger sizes to feature fusion to more effectively detect multi-scale objects, especially small objects. In addition, we reduced the number of convolutional channels in the neck to decrease the number of parameters required by the network. Based on these works, we obtained an improved feature pyramid network called multi-feature fusion pyramid network (MF-FPN). Finally, we modified the bounding box regression loss to enable the model to more accurately regress bounding boxes. This modification allows high-quality anchors to contribute more gradients to the training process. Therefore, the model can achieve better detection accuracy and faster convergence speed.

In summary, the contributions of this study are as follows:

1.  We propose a multi-scale object detector based on coordinate and global information aggregation for UAV aerial images, which can better detect targets with obscure features and targets with different scales;
2.  To alleviate the problem of the non-apparent object features due to occlusion and low light, the CGAM is proposed. The module can capture local, coordinate, and global contextual information and fuse them to reduce the interference of background factors on the feature extraction process, thereby obtaining more robust feature information;
3.  To make the model better detect multi-scale targets, the FFM is proposed. The module can learn the importance of different scale features in fusion and improve the representation ability of multi-scale features by reusing the features of different scales to improve the ability of model detection of multi-scale targets. At the same time, a larger size feature map is added to the feature fusion structure so that the model can detect small targets better. We named the improved feature pyramid network MF-FPN;
4.  To more accurately regress the bounding boxes, we modified the bounding box loss to improve the positioning effect of bounding boxes and make the model converge faster;
5.  We validated our CGMDet on two public UAV image datasets. The experimental results show that our model can better detect multi-scale targets and targets with less obvious features in UAV images.

## 2. Related Work

### 2.1. Object Detection

At present, the commonly used detectors are one-stage and two-stage detectors. Among them, the first step of the two-stage detector is to generate candidate regions, and the second step is to classify and regress each candidate region. The one-stage detector performs classification and regression directly. Classic two-stage detectors include R-CNN [7], Fast R-CNN [8], Faster R-CNN [9], SPP-Net [10], etc. The accuracy of the two-stage detector is higher than that of the one-stage detector, but the speed is slower than the one-stage detector. Commonly used one-stage detectors include SSD [11], RetinaNet [12], YOLO series [13–18], etc. Recently, some anchor-free detectors have been invented. The anchor-free method uses the features of object centers or key points to replace the complex anchor design. For example, FCOS [19] treats each pixel on the feature map as a training sample and uses a four-dimensional vector to regress the predicted box. CenterNet [20] represents objects using their center points and predicts bounding boxes by predicting the offset of the center point and the width and height of the object. The above detector has achieved good results in natural images. However, for UAV images, existing detection methods still face significant challenges. To date, many object detection methods for UAV

images have been proposed. For example, Liu et al. [21] proposed an anchor-free detector, Edge YOLO, which can be detected in real-time on the edge computing platform. The data enhancement method is used to suppress overfitting in the training process, and a mixed random loss function is used to improve the detection accuracy of small targets. Jiang et al. [22] used the YOLO model to realize the transfer detection from ground thermal infrared video images to UAV thermal infrared videos. Aiming at the challenges of scale and the sparsity of object detection in aerial images, ClusDet [23] proposed that the CPNet is used to generate the clustering region of objects first and ScaleNet to predict the scale information for adjusting the clustering region. DMNet [24] determines the target area to be clipped through the generated density map and then fuses the detection results of the clipped area and the original image to obtain more accurate detection effects.

### 2.2. Attention Mechanism

Attention mechanisms are widely used because they allow the model to focus more on important information and ignore unimportant ones. The attention mechanism has played a significant role in object detection. The squeeze-and-excitation block (SE) [25] is a classic channel attention mechanism that can apply a weight to each feature channel, allowing the model to focus more on the important channel information. However, to save computation, SE performs a squeeze operation during the processing which can result in the loss of some channel information. To avoid losing channel information, the Efficient Channel Attention Module (ECA) [26] and Effective Squeeze-and-Excitation Block (ESE Block) [27] were proposed. To prevent channel information loss, the ECA uses a 1D convolution operation instead of the two fully connected layers in SE. The ESE Block removes the squeeze operation and uses one fully connected layer instead of the two fully connected layers in SE. In addition to channel attention mechanisms, coordinate attention (CA) [28] obtains feature maps integrated with spatial coordinate information by performing adaptive average pooling algorithms along the x and y directions of the feature map, respectively. Furthermore, the CBAM [29] introduces channel and spatial attention mechanisms to allocate different weights for different channels and spatial regions to obtain highly responsive feature information and improve the network performance. The adaptive attention fusion mechanism (AAFM) [30] adaptively fuses features within and between modules with learnable fusion factors to improve the feature representation.

To better extract contextual feature information and alleviate the interference caused by occlusion and weak light, we designed the CGAM to extract the local feature and continuously focus on the coordinate information. In addition, it combines global information to obtain features with richer contextual information.

### 2.3. Multi-Scale Feature Fusion

Targets in unmanned aerial vehicle (UAV) images have characteristics such as large-scale variations and a high proportion of small objects, which pose significant challenges to object detection tasks. In deep networks, low-level feature maps typically contain rich positional information, while high-level features typically contain rich semantic information. Therefore, for better performance, low-level and high-level features are usually fused. FPN [31] fuses the features of adjacent scales through a top–down path and horizontal connections. PANet [32] proposes a bidirectional fusion structure that combines features in both top–down and bottom–up directions. Zhao et al. [33] proposed MLFPN to extract more representative multi-level and multi-scale features through the TUM and FFM, and then integrate features through the SFAM to obtain features with rich contextual information. Tan et al. [34] proposed BiFPN, which adds lateral skip connections to the top–down and bottom–up pathways and assigns a learnable weight to each feature map during fusion to emphasize the importance of different feature maps for better feature fusion.

To improve the detection performance of multi-scale targets, we designed the FFM to learn the importance of different features in the fusion and improve the representation

ability of different scale features through the repeated use of features of different scales to improve the model's ability to detect multi-scale targets.

## 3. Methods

This paper proposes a multi-scale object detector based on coordinate and global information aggregation for UAV aerial images, named CGMDet. Firstly, we designed the CGAM, which allows the backbone to focus more on the coordinate information and global context information during the feature extraction process, to enhance the ability of the network to extract features. Then, the FFM was proposed to better fuse multi-scale features. Finally, we modified the bounding box loss to obtain better detection results. Figure 1 shows the overall architecture of CGMDet. Our CGMDet enables the better detection of targets under occluded and low-light conditions and multi-scale targets.

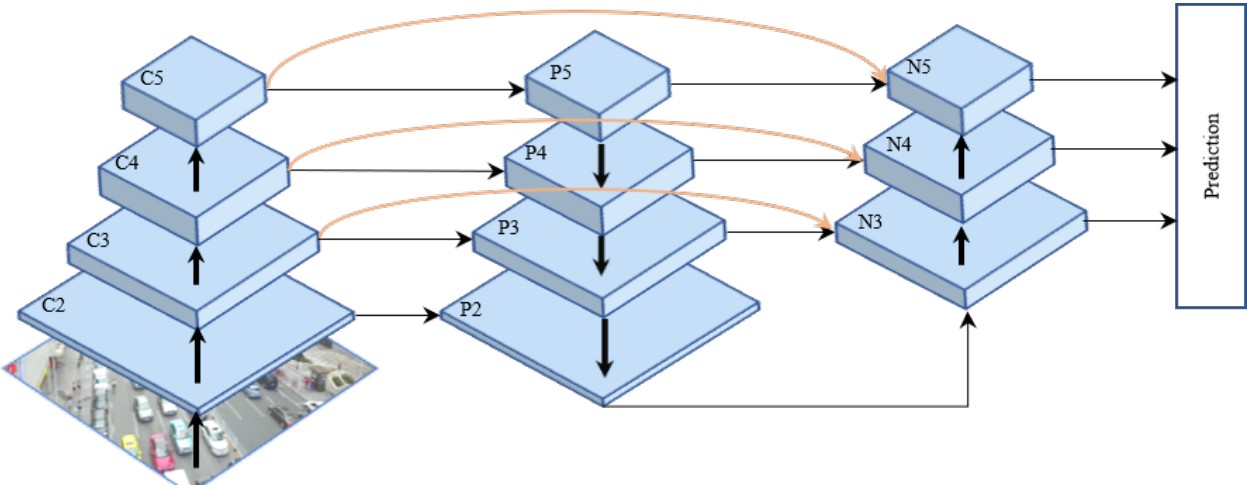

**Figure 1.** The overall architecture of CGMDet.

First, a $640 \times 640$ image is input into the backbone to extract features. The image passes through four CBS modules, which consist of convolution, batch normalization [35], and SiLU [36] activation function. In the first and third CBS modules, the convolution has a stride of 1, while in the second and fourth modules, it has a stride of 2. After obtaining the feature maps of four-fold downsampling, the CGAM extracts the feature. After that, the features of three scales were extracted by MP and CGAM. The MP module uses maximum pooling with the stride of 2 and $3 \times 3$ convolution with the stride of 2 to realize downsampling. SPPCSPC [18] aims to aggregate features with different receptive fields to obtain richer semantic contextual information. Then, the different scales of feature maps are fed into the neck, where our proposed FFM is used to fuse the features of different scales. Because most targets in UAV images are small, and low-level features are better for small target detection, we also input feature maps of size $160 \times 160$ into the feature fusion. Therefore, the feature map sizes that need to be fused in the neck are $160 \times 160$, $80 \times 80$, $40 \times 40$, and $20 \times 20$. ELAN-H [37] refined the fused features. RepConv can decouple the training and inference process, allowing the model to learn more knowledge during the training process without affecting the inference speed. Finally, the feature maps of size $80 \times 80$, $40 \times 40$, and $20 \times 20$ are used for object detection. Figure 2 shows the detailed network structure of our CGMDet.

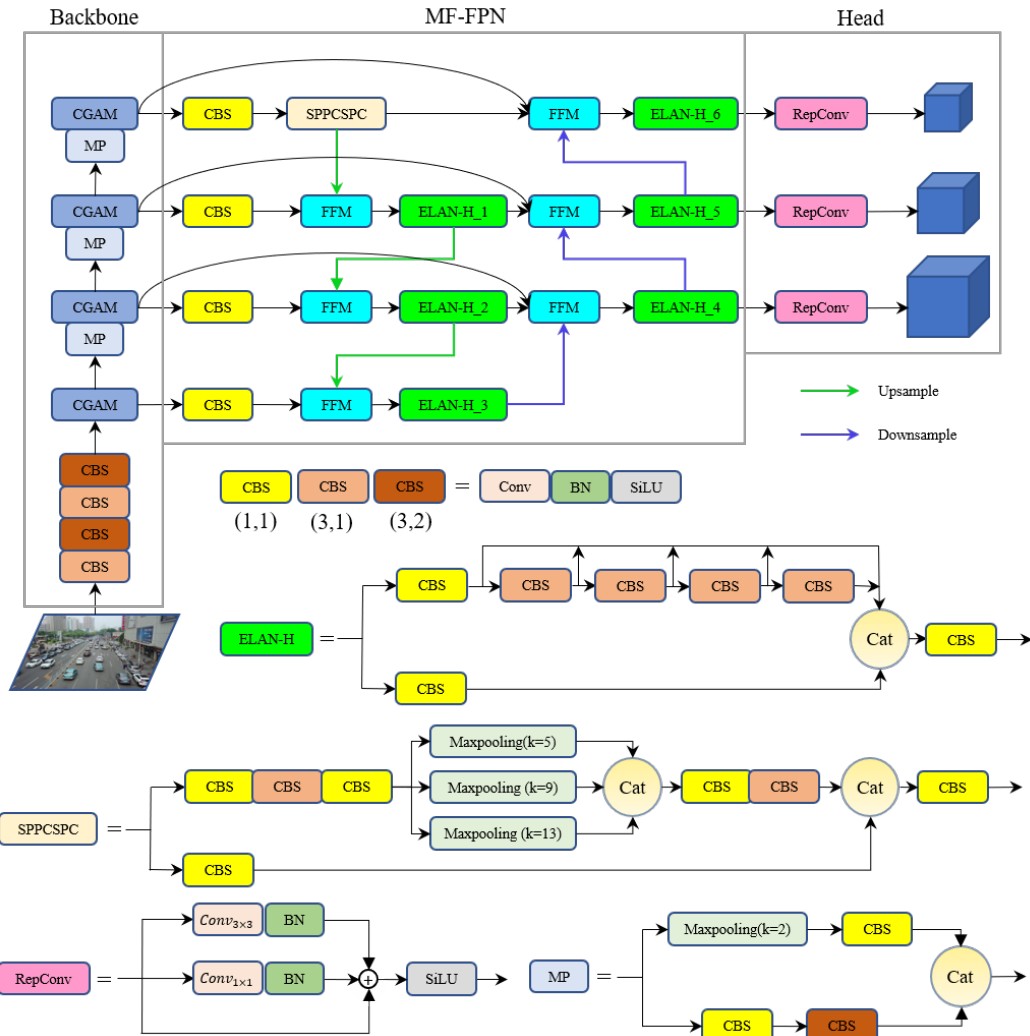

**Figure 2.** The detailed architecture of CGMDet.

### 3.1. Coordinate and Global Information Aggregation Module

Because of the camera angle of the UAV, there are often many cases where the targets are occluded. At the same time, the light is dimmer in nighttime scenes. The feature extraction process in these two scenarios is easily disturbed by background factors, which is easy to cause missed detections and false detections. To alleviate the interference of background factors, we designed the CGAM, which can integrate global information, coordinate information, and local information extracted by convolution, to obtain more robust features. The structure of CGAM is shown in Figure 3.

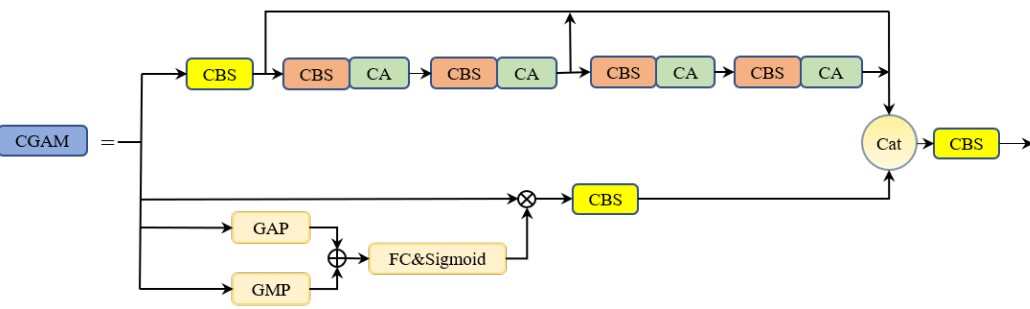

**Figure 3.** Coordinate and Global Information Aggregation Module.

The CGAM consists of two branches. The first branch introduces the coordinate attention mechanism, which constantly focuses on the coordinate information when using convolution for feature extraction. The second branch obtains global information on the feature map through two pooling operations. By fusing the features extracted from the two branches, richer contextual features are obtained.

The first branch of CGAM first uses a $1 \times 1$ convolution to reduce the number of channels in the input feature $X \in \mathbb{R}^{C \times H \times W}$ by half, obtaining the first intermediate feature map $M_1 \in \mathbb{R}^{\frac{C}{2} \times H \times W}$. As shown in Formula (1):

$$M_1 = Conv_{1 \times 1}(F) \tag{1}$$

The features map is then extracted using $3 \times 3$ convolution and the coordinate attention mechanism, obtaining the second and third intermediate output feature maps $M_2$, $M_3 \in \mathbb{R}^{\frac{C}{2} \times H \times W}$. As shown in Formulas (2) and (3):

$$M_2 = CA(Conv_{3 \times 3}(CA(Conv_{3 \times 3}(M_1)))) \tag{2}$$

$$M_3 = CA(Conv_{3 \times 3}(CA(Conv_{3 \times 3}(M_2)))) \tag{3}$$

where CA represents the coordinate attention mechanism. The coordinate attention is shown in Figure 4.

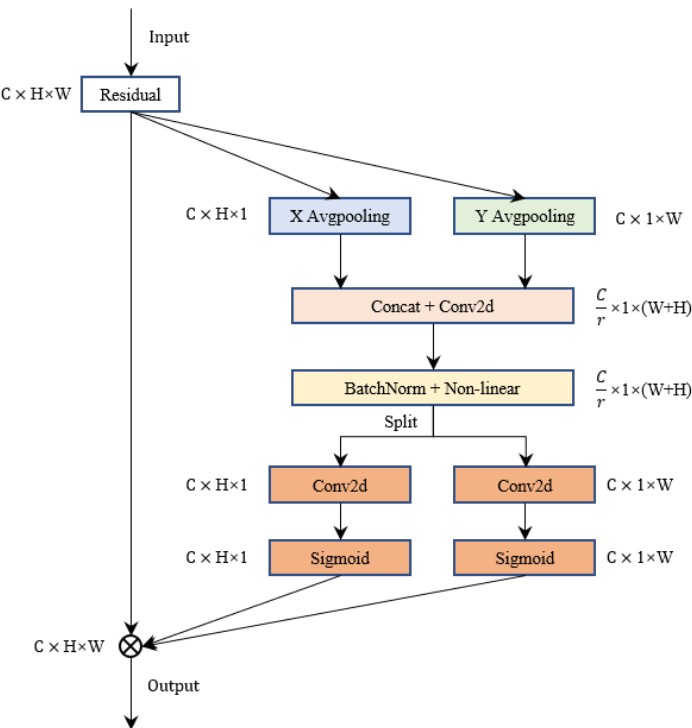

**Figure 4.** Coordinate Attention.

The coordinate attention mechanism first performs pooling operations on the input feature $F \in \mathbb{R}^{C \times H \times W}$ along the horizontal and vertical directions, obtaining features $f^h \in \mathbb{R}^{C \times H \times 1}$ and $f^w \in \mathbb{R}^{C \times 1 \times W}$. As shown in Formulas (4) and (5):

$$f_c^h = \frac{1}{W} \sum_{0 \leq i \leq W} F_c(h, i) \tag{4}$$

$$f_c^w = \frac{1}{H} \sum_{0 \leq j \leq H} F_c(j, w) \tag{5}$$

where $F_c$ and $f_c$ represent the $c$-th channel of the input and output features, respectively. $W$ and $H$ represent the width and height of the input feature, respectively. Then, $f^h$ and $f^w$ are concatenated along the spatial dimension, and the number of channels is reduced using a $1 \times 1$ convolution. Furthermore, feature $Q \in \mathbb{R}^{\frac{C}{r} \times 1 \times (W+H)}$ is obtained by passing it through batch normalization and an activation function, where $r$ is a scaling factor. Batch normalization is used to prevent gradient explosion or vanishing, making the model more stable during training, and the activation function introduces nonlinear factors to enhance the expression ability of the model. The formula is shown as (6):

$$Q = \delta\left(BN\left(Conv_{1\times1}\left(\left[f^h, f^w\right]\right)\right)\right) \tag{6}$$

where $[\cdot]$ denotes the channel concatenation operation. $BN$ denotes batch normalization. $\delta$ represents a nonlinear activation function. Then, the feature tensor $Q$ is split along the spatial dimension to obtain two feature tensors $y^h \in \mathbb{R}^{\frac{C}{r} \times H \times 1}$ and $y^w \in \mathbb{R}^{\frac{C}{r} \times 1 \times W}$. Increasing the number of channels for $y^h$ and $y^w$ to the same as the input feature map $F$ by $1 \times 1$ convolution, and then the attention weights $g^h$ and $g^w$ are obtained by the sigmoid function. The formulas are shown as (7) and (8):

$$g^h = \sigma(Conv_{1\times1}(y^h)) \tag{7}$$

$$g^w = \sigma(Conv_{1\times1}(y^w)) \tag{8}$$

where $\sigma$ denotes the sigmoid function. Finally, $g^h$ and $g^w$ are multiplied by the feature map $F$. As shown in Formula (9):

$$CA = F \otimes g^h \otimes g^w \tag{9}$$

The second branch of the CGAM module first uses global pooling operations to add the global contextual information of the backbone network. For the input feature $X \in \mathbb{R}^{C \times H \times W}$, perform global average pooling and global maximum pooling operations first, then add the results, and finally allocate weights for each channel through a fully connected layer and a sigmoid function, making the model focus on the highly responsive channel information. As shown in Formula (10):

$$O = \sigma(FC(GAP(X) \oplus GMP(X))) \tag{10}$$

where $GAP$ and $GMP$ represent global average pooling and global maximum pooling, respectively. $\sigma$ represents the sigmoid function. $FC$ represents a fully connected layer. Then, multiply the result with the input feature $X$ and use a $1 \times 1$ convolution to obtain the output $M_4 \in \mathbb{R}^{\frac{C}{2} \times H \times W}$ of the second branch. As shown in Formula (11):

$$M_4 = Conv_{1\times1}(X \otimes O) \tag{11}$$

The CGAM module first performs channel concatenation on all intermediate output features $M_1$, $M_2$, $M_3$, and $M_4$ from the two branches, and then uses a $1 \times 1$ convolution to obtain the output feature $Z \in \mathbb{R}^{2C \times H \times W}$ of CGAM. As shown in Formula (12):

$$Z = Conv_{1\times1}([M_1, M_2, M_3, M_4]) \tag{12}$$

Our CGAM module can extract coordinate information, global information, and local information simultaneously and fuse them to obtain more robust features, thereby accurately locating the target, reducing the focus of the model on the background, and improving the detection ability of the model.

### 3.2. Multi-Feature Fusion Pyramid Network

The object scale changes greatly in UAV images, and there are many small objects. To enhance the ability of the network to detect multi-scale targets, we proposed the FFM, which can learn the importance of different features in fusion to integrate the features of different scales better. Moreover, reusing feature maps to enrich the context information of fused features and improve the expression ability of multi-scale features. In this way, the model can improve the detection ability of multi-scale targets. We added the feature map of size $160 \times 160$ to the MF-FPN for fusion to alleviate the difficulty in detecting small objects. The FFM is shown in Figure 5.

There are two main ways to fuse features. If only two feature maps need to be fused, such as the top–down path in the neck, the method shown in Figure 5a is used. This method assigns two learnable weights to the two feature maps to determine the importance of each feature map, as shown in Formula (13):

$$P = \frac{w_1 F_1 + w_2 F_2}{w_1 + w_2 + \Delta} \tag{13}$$

where $w_1$ and $w_2$ are learnable parameters, and $\Delta$ is a small number to avoid numerical instability. For the case of fusing three feature maps, we use the method shown in Figure 5b. The calculation method is shown in Formula (14). We first use Formula (13) to fuse the three feature maps pairwise, where each feature map participates in two fusions, achieving the effect of reusing features. Then, we obtain three different intermediate feature maps and finally assign three learnable weights to fuse these three feature maps, obtaining the output feature map with rich contextual information for the final prediction.

$$N = \frac{w_1 P(F_1, F_2) + w_2 P(F_1, F_3) + w_3 P(F_2, F_3)}{w_1 + w_2 + w_3 + \Delta} \tag{14}$$

Since the fusion features contain feature maps with different scales and channel numbers, it is necessary to adjust the size and number of channels of the feature maps to be consistent before fusion. Our FFM integrates multi-scale features better by automatically learning the importance of different scale feature maps in the fusion process. Furthermore, the feature maps of different scales can be used to enrich the context information of fusion features and improve the representation ability of multi-scale features to enhance the detector's perception of targets of different scales.

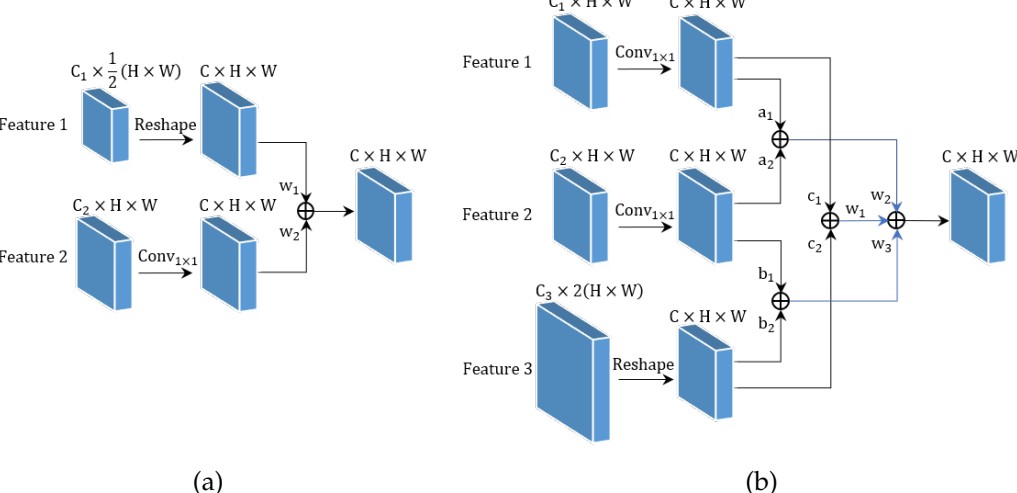

(a)    (b)

**Figure 5.** The structure of the Feature Fusion Module (FFM). (**a**) represents the feature fusion method in the top-down path; (**b**) represents the feature fusion method in the bottom-up path.

To preserve more feature information, the convolutional channels in the model are usually large, and larger channels bring more parameters to the model. The number of parameters of convolution can be calculated by the Formula (15).

$$Params = K_h \times K_w \times C_{in} \times C_{out} \tag{15}$$

where $K_w$ and $K_h$ represent the size of the convolutional kernel. $C_{in}$ and $C_{out}$ represent the number of input and output channels of the convolution, respectively. Therefore, to decrease the parameters of the model, we modified the convolutional channel numbers in the neck of the model. First, the channel numbers of the $3 \times 3$ convolutions in ELAN-H_1, ELAN-H_2, and ELAN-H_3 to 32. Then, we adjusted the output channel numbers of the first two $1 \times 1$ convolutions in ELAN-H_4, ELAN-H_5, and ELAN-H_6 to 1/4 of the input channel numbers. ELAN-H_1, ELAN-H_2, ELAN-H_3, ELAN-H_4, ELAN-H_5, and ELAN-H_6 have the same structure. The changes in the number of parameters for all ELAN-H modules in the neck are shown in Table 1.

**Table 1.** Change of parameters in the neck part of the model.

| Module | Baseline | Ours |
|---|---|---|
| ELAN-H_1 | 1.26 M | 0.1 M |
| ELAN-H_2 | 0.32 M | 0.07 M |
| ELAN-H_3 | \ | 0.07 M |
| ELAN-H_4 | \ | 0.21 M |
| ELAN-H_5 | 1.26 M | 0.85 M |
| ELAN-H_6 | 5.05 M | 3.4 M |

After the above improvements, we named the improved feature pyramid structure MF-FPN. The pseudo-code of the MF-FPN is shown in Algorithm 1. We first fuse two adjacent features in $X = \{x_1, x_2, x_3, x_4\}$ from top to bottom to obtain four intermediate feature maps $M = \{m_1, m_2, m_3, m_4\}$. Then, the features from $X$ and $M$ are fused using the bottom–up path and skip connections to obtain three final features of different scales, denoted as $Y = \{y_1, y_2, y_3\}$, which will be used for prediction.

---

**Algorithm 1** The feature fusion method of MF-FPN.

---

**Input: X** $= \{x_1, x_2, x_3, x_4\}$, $X$ refers to four different scale feature maps of the backbone network output. The scale of $x_1$ is the smallest and $x_4$ is the largest.

**Step 1: M** $= \{\}$, **M** refers to the intermediate feature map generated by the top–down branch of MF-FPN. $Conv()$ represents a series of convolution operations required, $Reshape()$ represents the upsampling and downsampling operations required, and $FFM()$ represents our feature fusion operation.

    **for** $i = 1$ to 4 **do**
        **if** $i = 1$ **then**
            $m_i = Conv(x_i)$
        **else**
            $m_i = Conv(FFM(x_i, Reshape(x_{i-1})))$
        **end if**
        **M**.$append(m_i)$
    **end for**

**Step 2: Y** $= \{\}$, **Y** refers to the feature map generated by the bottom–up branch of MF-FPN for use in prediction.

    **for** $i = 1$ to 3 **do**
        $y_i = Conv(FFM(x_i, M_i, Reshape(M_{i+1})))$
        **Y**.$append(y_i)$
    **end for**

**Output:** Return **Y**.

---

### 3.3. Loss Function

The CIOU [38] loss is commonly used as the bounding box regression loss in existing models. The definition of CIOU is as follows:

$$L_{CIOU} = 1 - IOU + \frac{\rho^2(b, b^{gt})}{c^2} + \alpha v \tag{16}$$

$$IOU = \frac{|B \cap B^{gt}|}{|B \cup B^{gt}|} \tag{17}$$

where $\rho$ represents the Euclidean distance. $b$ and $b^{gt}$ denote the center points of the predicted and ground truth box, respectively. $c$ represents the diagonal length of the minimum bounding rectangle of the ground truth box and predicted box. $v$ and $\alpha$ are defined as follows:

$$v = \frac{4}{\pi^2} \left( arctan \frac{w^{gt}}{h^{gt}} - arctan \frac{w}{h} \right)^2 \tag{18}$$

$$\alpha = \frac{v}{(1 - IOU) + v} \tag{19}$$

where $h^{gt}$ and $w^{gt}$ are the height and width of the ground truth box. $h$ and $w$ are the height and width of the predicted box.

However, accurately regressing the height and width of the bounding box cannot only be achieved through the aspect ratio. Because when $w = kw^{gt}$ and $h = kh^{gt}(k \in R^+)$, $v = 0$. The EIOU loss [39] not only retains the advantages of the CIOU but also minimizes the differences between the height and width of the predicted and ground truth boxes, resulting in better localization performance. The definition of EIOU is shown in Formula (20).

$$L_{EIOU} = 1 - IOU + \frac{\rho^2(b, b^{gt})}{c^2} + \frac{\rho^2(h, h^{gt})}{h^c} + \frac{\rho^2(w, w^{gt})}{w^c} \tag{20}$$

where $h^c$ and $w^c$ are the height and width of the minimum bounding box surrounding the ground truth and predicted box. EIOU can directly regress the height and width of the prediction box. Furthermore, to make the model converge faster, we use the focal EIOU loss [39], which combines the focal loss with the EIOU loss, as the bounding box regression loss for CGMDet. It allows high-quality anchors to contribute more gradients to the training process, thereby improving the convergence speed of the model. Its definition is as follows:

$$L_{Focal-EIOU} = IOU^\lambda \times L_{EIOU} \tag{21}$$

where $\lambda$ is an adjustable parameter, which we set to 0.5. Additionally, the confidence and classification losses of the model are calculated using binary cross entropy with Logits Loss (BCEWithLogitsLoss) [40]. The definition is as follows:

$$L_{BCE} = - \sum_{n=1}^{N} \hat{y}_i \log(\sigma(y)) + (1 - \hat{y}_i) \log(\sigma(1 - y)) \tag{22}$$

where $N$ is the number of input vectors. $\hat{y}_i$ and $y$ are the predicted and truth vectors, respectively. $\sigma$ is the sigmoid function. The overall loss of CGMDet can be obtained by combining the classification loss, confidence loss, and bounding box regression loss. The definition is as follows:

$$\begin{aligned} Loss &= \lambda_1 L_{box} + \lambda_2 L_{obj} + \lambda_3 L_{cls} \\ &= \lambda_1 L_{Focal-EIOU}(P_{box}, T_{box}) + \lambda_2 L_{BCE}(P_{obj}, T_{obj}) + \lambda_3 L_{BCE}(P_{cls}, T_{cls}) \end{aligned} \tag{23}$$

where $L_{box}$, $L_{obj}$, and $L_{cls}$ represent the bounding box regression loss, confidence loss, and classification loss, respectively. $P_{box}$ and $T_{box}$ denote the predicted and ground truth box, respectively. $P_{obj}$ and $T_{obj}$ denote the predicted and truth confidence, respectively. Furthermore, $P_{cls}$ and $T_{cls}$ represent the predicted and truth class probability, respectively. The hyperparameters $\lambda_1$, $\lambda_2$, and $\lambda_3$ are set to 0.05, 0.7, and 0.3 by default.

## 4. Experiments

To verify the effectiveness of our detector, we conducted experiments on two publicly available UAV image datasets and compared it with other detectors.

### 4.1. Datasets

The VisDrone benchmark [41] contains 10,209 static images, among which 6471 images are used for training, 3190 for testing, and 548 for validation. The resolution of the images is approximately 2000 × 1500 pixels, collected by various drone platforms in different scenarios, as well as under different weather and lighting conditions. Figure 6 shows some images from this dataset.

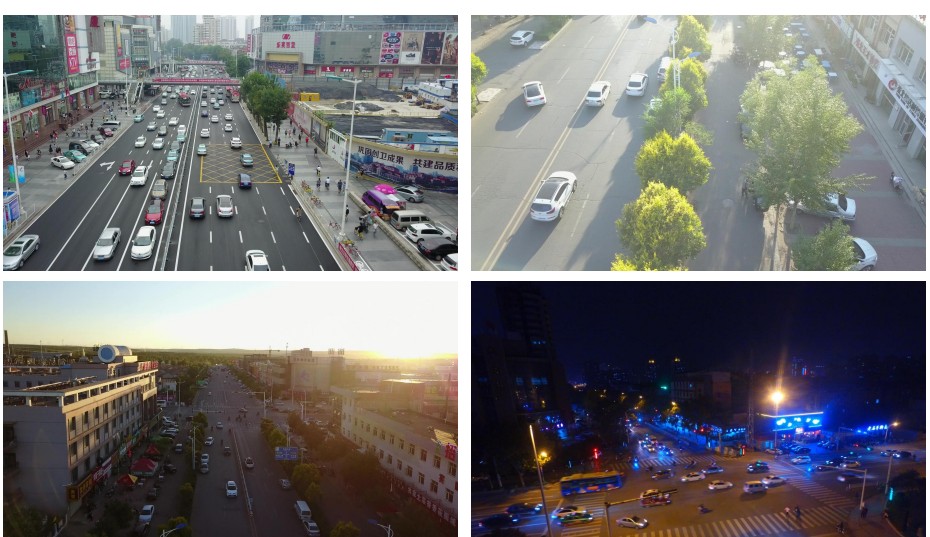

**Figure 6.** Some examples in VisDrone.

The UAVDT benchmark [42] consists of 40,735 images, among which 24,206 images are used for training and 16,529 for validation. This dataset contains images with different weathers, flight heights, shooting angles, and occlusion scenes. The images in the dataset have a resolution of approximately 1080 × 540 pixels. The dataset includes three predefined classes: car, truck, and bus. Figure 7 shows some images from this dataset.

### 4.2. Implementation and Evaluation Criteria

4.2.1. Implementation

This paper validated the proposed object detector on the Ubuntu 18.04.6 LTS system, trained and tested on NVIDIA GeForce RTX 3090 (24 G) as the graphics processing unit, with an Intel(R) Xeon(R) Silver 4114 CPU @2.20 GHz and Python version 3.6. The CUDA version used was 11.7 and the PyTorch version used was 1.10.2.

During model training, the input image size was set to 640 × 640, and the Stochastic Gradient Descent (SGD) optimizer with momentum was used. The initial learning rate was set to 0.01, the momentum parameter was set to 0.937, the weight decay coefficient was set to 0.0005, and the batch size was set to 8. The total number of training iterations for the VisDrone dataset was 300, and for the UAVDT dataset was 200.

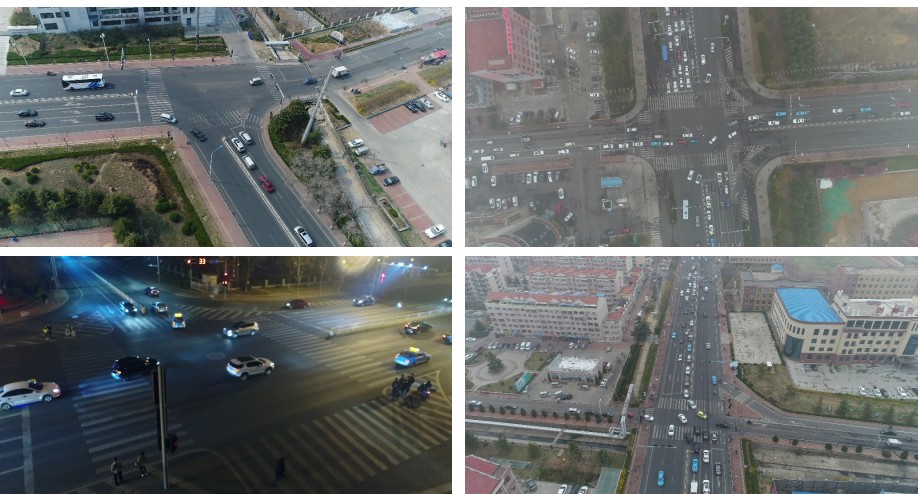

**Figure 7.** Some examples in UAVDT.

### 4.2.2. Evaluation Criteria

Precision *P*, recall *R*, average precision *AP*, and mean average precision *mAP* are used as metrics to evaluate the performance of our detector. *P* represents how many predicted positive samples are correct. *R* represents how many positive samples are predicted. The definitions of *P* and *R* are as follows:

$$P = \frac{TP}{TP + FP} \tag{24}$$

$$R = \frac{TP}{TP + FN} \tag{25}$$

where *TP* represents how many samples were correctly predicted to be positive. *FP* represents how many samples were incorrectly predicted to be positive. *FN* represents how many samples were incorrectly predicted to be negative. *P* and *R* are usually trade-offs between each other. Therefore, *AP* can better measure the detection capability of the network. The definition of *AP* is shown in Formula (26):

$$AP = \int_0^1 P(R)dR \tag{26}$$

where $P(R)$ represents the precision value when the recall value on the P-R curve is *R*. *mAP* is the average of all class *AP* values, which can represent the average detection performance of the detector on the dataset. The definition is shown in Formula (27):

$$mAP = \frac{1}{K} \sum_{i=1}^{K} AP_i \tag{27}$$

where $AP_i$ denotes the *AP* value of the *i*-th category. *K* denotes the number of target categories.

To better describe the ability of our detector to detect multi-scale objects, we also used COCO evaluation metrics [43], such as $AP_S$, $AP_M$, and $AP_L$. $AP_S$ represents the *AP* value of small objects with an area less than $32 \times 32$. $AP_M$ represents the *AP* value of medium objects with an area between $32 \times 32$ and $96 \times 96$. $AP_L$ represents the *AP* value of large objects with an area greater than $96 \times 96$.

### 4.3. Experimental Results

#### 4.3.1. Experimental Results on the VisDrone Dataset

We evaluated our model using the VisDrone dataset and compared our model with other models to validate its effectiveness. As shown in Table 2, our CGMDet achieved the highest mAP0.5 and mAP. Compared with baseline YOLOv7 [18], mAP0.5 and mAP are

increased by 1.9% and 1.2%, respectively. At the same time, CGMDet achieved the second-highest result on mAP0.75 and $AP_M$, where it was only 0.1% lower than CDMNet's $AP_M$. Compared with YOLOv7, our CGMDet increases by 1.3% $AP_S$ and 1.2% $AP_M$. Although the value of $AP_L$ is 0.4% lower than that of YOLOv7, on the whole, CGMDet is superior to YOLOv7 in the detection performance of multi-scale targets, which also indicates that our proposed FFM can better integrate the features of different scales and improve the model's perception ability of features of different scales. Compared to Edge YOLO, our CGMDet is 5.7% lower on $AP_L$, but our CGMDet is better than Edge YOLO on other metrics. Compared with NWD, CGMDet is 10.6% higher at mAP0.5 but 2.0% lower at $AP_S$. Although our CGMDet is lower than ClusDet and DMNet on $AP_L$, it is higher than them on other metrics. Our CGMDet was only 0.2% higher than CEASC on mAP0.5, but significantly higher than CEASC on mAP0.75 and mAP. At the same time, compared with CDMNet, CGMDet has slight disadvantages in mAP0.75, $AP_S$, and $AP_M$, but has obvious advantages in mAP0.5 and $AP_L$. Compared to RetinaNet, Cascade-RCNN, Faster-RCNN, YOLOv3, YOLOX, YOLOv5l, HawkNet, and QueryDet, our CGMDet outperformed them on all metrics.

**Table 2.** Comparison with state-of-the-art detectors on the VisDrone dataset.

| Method | mAP0.5 | mAP0.75 | mAP | $AP_S$ | $AP_M$ | $AP_L$ |
|---|---|---|---|---|---|---|
| RetinaNet [12] | 35.9 | 18.5 | 19.4 | 14.1 | 29.5 | 33.7 |
| Cascade R-CNN [44] | 39.9 | 23.4 | 23.2 | 16.5 | 36.8 | 39.4 |
| Faster R-CNN [9] | 40.0 | 20.6 | 21.5 | 15.4 | 34.6 | 37.1 |
| YOLOv3 [15] | 31.4 | 15.3 | 16.4 | 8.3 | 26.7 | 36.9 |
| YOLOX [45] | 45.0 | 26.6 | 26.7 | 17.4 | 37.9 | 45.3 |
| YOLOv5l [17] | 36.2 | 20.1 | 20.5 | 12.4 | 29.9 | 36.4 |
| HawkNet [46] | 44.3 | 25.8 | 25.6 | 19.9 | 36.0 | 39.1 |
| QueryDet [47] | 48.1 | 28.8 | 28.3 | \ | \ | \ |
| Edge YOLO [21] | 44.8 | 26.2 | 26.4 | 16.3 | 38.7 | 53.1 |
| NWD [48] | 40.3 | \ | \ | **22.2** | \ | \ |
| ClusDet [23] | 50.6 | 24.7 | 26.7 | 17.6 | 38.9 | 51.4 |
| DMNet [24] | 47.6 | 28.9 | 28.2 | 19.9 | 39.6 | **55.8** |
| CEASC [49] | 50.7 | 28.4 | 28.7 | \ | \ | \ |
| CDMNet [50] | 49.5 | **29.8** | 29.2 | 20.8 | **40.7** | 41.6 |
| YOLOv7 [18] | 49.0 | 27.8 | 28.1 | 18.9 | 39.4 | 47.8 |
| CGMDet(Ours) | **50.9** | 29.4 | **29.3** | 20.2 | 40.6 | 47.4 |

We also listed the mAP0.5 for each category to describe in more detail which categories our model has improved on. As shown in Table 3, our model has a higher mAP0.5 than other models for each category. In addition, except for the tricycle category, which has the same result as YOLOv7, all other categories have greatly improved, especially the bus and bicycle categories, which increased by 3.3% and 3.8%, respectively.

**Table 3.** Detection results for each category on the VisDrone dataset.

| Method | Pedestrian | People | Bicycle | Car | Van | Truck | Tricycle | Awing-Tricycle | Bus | Motor | mAP0.5 |
|---|---|---|---|---|---|---|---|---|---|---|---|
| YOLOv3 [15] | 12.8 | 7.8 | 4.0 | 43.0 | 23.5 | 16.5 | 9.5 | 5.1 | 29.0 | 12.5 | 31.4 |
| YOLOv5l [17] | 44.4 | 36.8 | 15.6 | 73.9 | 39.2 | 36.2 | 22.6 | 11.9 | 50.5 | 42.8 | 37.4 |
| YOLOv7 [18] | 57.6 | 48.7 | 21.6 | 85.4 | 51.9 | 45.8 | 37.9 | 18.3 | 63.0 | 60.0 | 49.0 |
| CGMDet (ours) | **59.7** | **50.7** | **25.4** | **86.2** | **53.4** | **47.4** | **37.9** | **20.2** | **66.3** | **61.6** | **50.9** |

To make it more deployable on mobile devices, we also designed a tiny version of the model and conducted experiments. The inference time was obtained by calculating the average prediction time of all images in the test set. The results in Table 4 show that our CGMDet-tiny achieved the best results. Compared to YOLOv7-tiny, our CGMDet-tiny

achieved a 4% improvement in mAP0.5, 3.6% in mAP0.75, and 3.1% in mAP. Meanwhile, $AP_S$, $AP_M$, and $AP_L$ increased by 2.8%, 4.1%, and 1%, respectively. Moreover, our model has only increased by 1.4 M parameters. Although our CGMDet-tiny is slower than other models in terms of inference time, it can still detect in real time. Moreover, judging from the results of detection performance, it is worth trading inference time for detection accuracy.

**Table 4.** Comparison of detection results of tiny version on VisDrone dataset.

| Method | mAP0.5 | mAP0.75 | mAP | $AP_S$ | $AP_M$ | $AP_L$ | Params (M) | Inference Time (ms) |
|---|---|---|---|---|---|---|---|---|
| YOLOX-tiny [45] | 35.7 | 19.2 | 19.7 | 12.2 | 28.3 | 31.7 | 5.0 | \ |
| YOLOv5s [17] | 28.7 | 14.0 | 15.1 | 9.2 | 22.2 | 31.8 | 7.0 | 10.8 |
| YOLO-UAVlite [51] | 36.6 | 19.7 | 20.6 | 12.9 | 29.3 | 33.4 | 1.4 | \ |
| YOLOv7-tiny [18] | 35.8 | 17.3 | 18.6 | 11.4 | 27.4 | 36.6 | 6.0 | 9.4 |
| CGMDet-tiny (ours) | **39.8** | **20.9** | **21.7** | **14.2** | **31.5** | **37.6** | 7.4 | 21.2 |

To better illustrate the advantages of our model, we provide the detection results of several images in different scenarios. As shown in Figure 8(a1–a4) are the results of YOLOv7, and Figure 8(b1–b4) are the detection results of our CGMDet. From the red dashed box in (a1,b1) of Figure 8, YOLOv7 recognized that text on the ground as a car, while our model can recognize it as the background. From Figure 8(a2,b2), our model can also distinguish between two objects with very similar features that are close together. Due to the indistinct features of small targets, it is difficult for the model to learn, and it is easy to recognize similar backgrounds as targets. However, our improved model is better able to detect small targets and can effectively distinguish the background, as shown in Figure 8(a3,b3). In addition, we also tested the detection performance in nighttime scenes, as shown in the red dashed box in Figure 8(a4,b4). YOLOv7 failed to detect it, while our model accurately marked it out.

As shown in Table 5, in order to verify the effectiveness of the proposed CGAM module, we compared it with other similar modules. Compared to ELAN, our CGAM improved the mAP0.5 by 0.7% and mAP by 0.5%. Although the mAP0.75 and mAP of CGAM have a slight disadvantage compared with the CSPRepResStage, the number of parameters brought to the model by CGAM is much smaller than that of the CSRepResStage. The results show that our CGAM can effectively enhance the model's feature extraction ability without too many parameters.

We also compared our MF-FPN with other multi-scale fusion methods. We only incorporate the last three scale features generated by the YOLOv7 backbone. As shown in Table 6, our MF-FPN is similar to PAFPN on all metrics, but MF-FPN has fewer parameters and computations than PAFPN. Although the number of parameters and the calculation amount are the lowest among the three, the BiFPN also has a gap with the other two in performance. The results show that our MF-FPN can effectively integrate multi-scale features and enhance the model's perception for multi-scale targets.

**Table 5.** Comparison between CGAM and similar methods.

| Method | mAP0.5 | mAP0.75 | mAP | Param (M) |
|---|---|---|---|---|
| YOLOv7 + ELAN [37] | 49.0 | 27.8 | 28.1 | 36.5 |
| YOLOv7 + CSPRepResStage [52] | 49.7 | **28.2** | **28.7** | 41.4 |
| YOLOv7 + C2f [53] | 48.0 | 26.8 | 27.5 | 36.7 |
| YOLOv7 + CGAM (ours) | **49.7** | 28.0 | 28.6 | 38.0 |

**Table 6.** Comparison of different multi-scale fusion methods.

| Method | mAP0.5 | mAP0.75 | mAP | Param (M) | GFLOPs |
|---|---|---|---|---|---|
| YOLOv7 + PAFPN | 49.0 | **27.8** | 28.1 | 36.53 | 103.3 |
| YOLOv7 + BiFPN | 48.6 | 27.2 | 27.9 | 35.97 | 102.5 |
| YOLOv7 + MF-FPN (ours) | **49.0** | 27.6 | **28.2** | 36.50 | 102.9 |

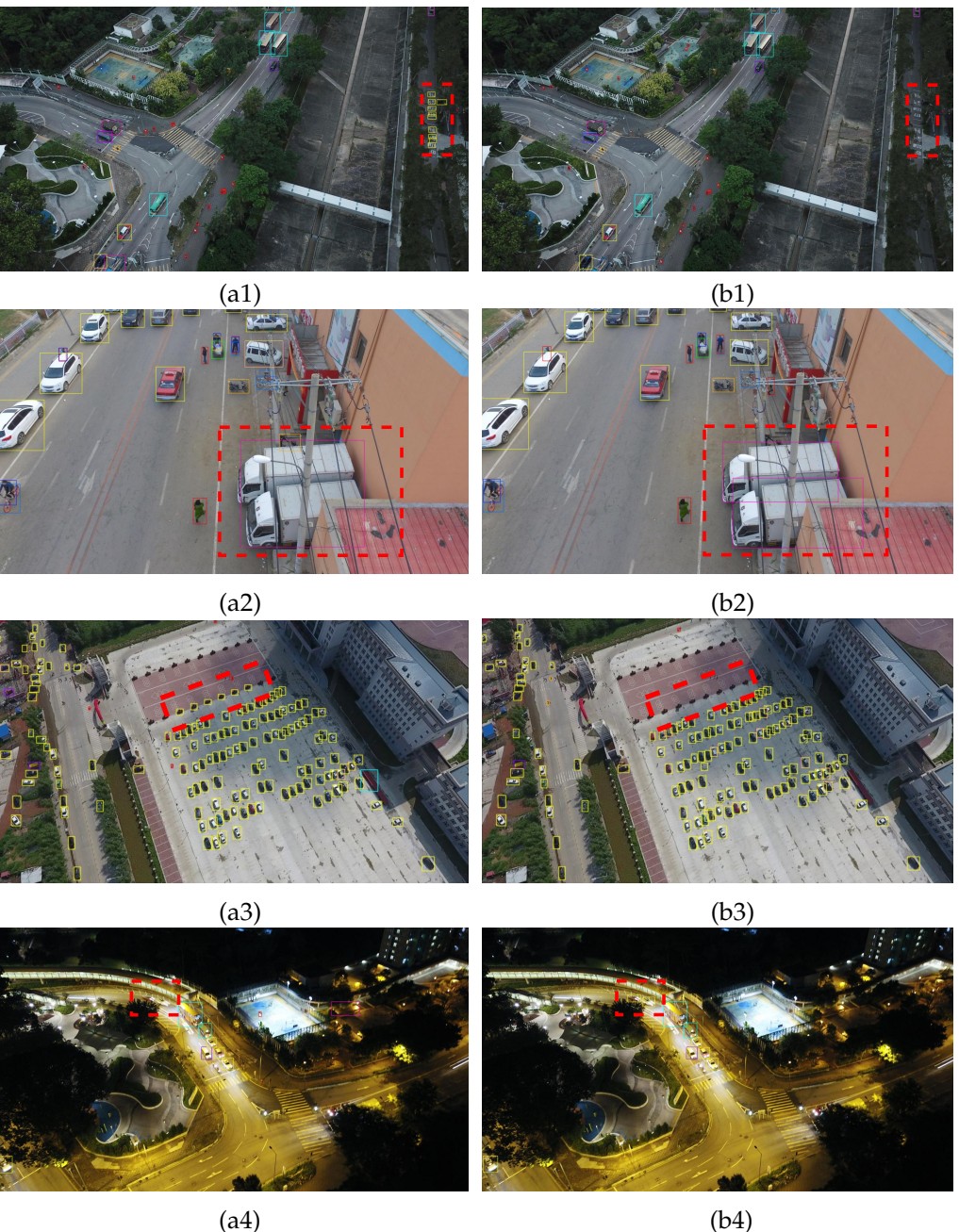

**Figure 8.** The detection results of the VisDrone dataset under different scenes. (**a1–a4**) are the detection results of YOLOv7; (**b1–b4**) are the results of the proposed model.

As shown in Figure 9, we present the visualized results of three different multi-scale fusion methods. As shown in Figure 9(b1), there are many false detections in the red dotted box, while the PAFPN and our MF-FPN reduce the number of false check targets. In the green dotted box, the MF-FPN accurately boxes the target, while the BiFPN and PAFPN

do not fully box the target. As shown in the red and green dotted boxes in Figure 9(b2), BiFPN missed one of the two adjacent targets, while neither PAFPN nor MF-FPN missed it, but MF-FPN's bounding box in the green dotted box was less accurate than PAFPN's. As shown in the green dotted boxes in Figure 9(a3,b3), although PAFPN and BiFPN detected the truck, they generated a redundant detection box, and the detection box's scope is not accurate. However, the MF-FPN does not generate redundant detection boxes, and the detection box can accurately enclose the target. It can also be seen from Figure 9 that our MF-FPN can also better detect targets of different scales.

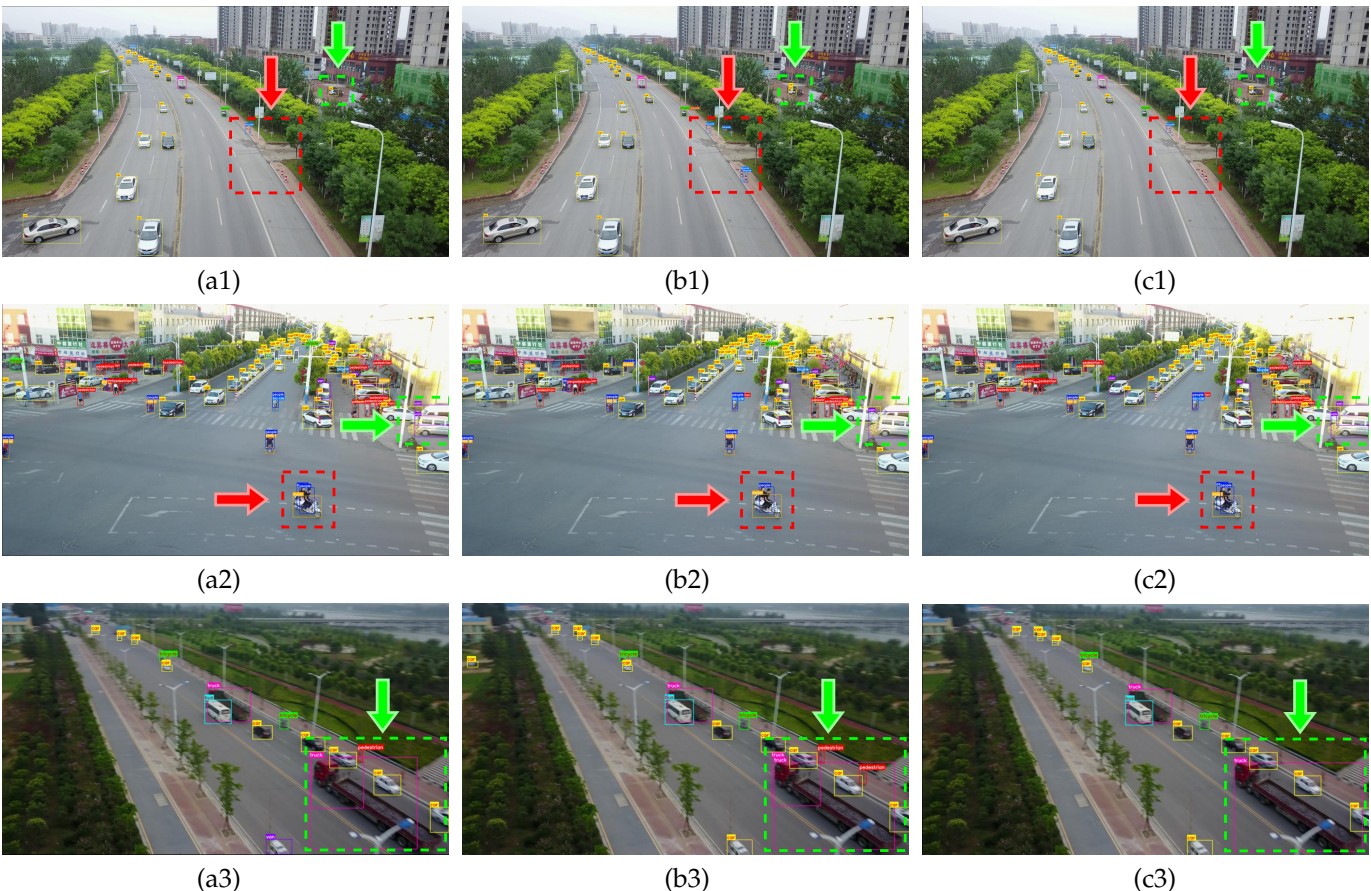

(a1)　　　　　　　　(b1)　　　　　　　　(c1)

(a2)　　　　　　　　(b2)　　　　　　　　(c2)

(a3)　　　　　　　　(b3)　　　　　　　　(c3)

**Figure 9.** Visual results of different fusion methods. (**a1**–**a3**) are the visualization results of PAFPN; (**b1**–**b3**) are the visual results of BiFPN; and (**c1**–**c3**) are visualization results for MF-FPN.

### 4.3.2. Experimental Results on the UAVDT Dataset

We also evaluated our model using the UAVDT dataset and compared our model with others. As shown in Table 7, compared with YOLOv7, our model has increased mAP0.5 by 3.0%, mAP0.75 by 3.2%, and mAP by 2.3%. The detection performance of YOLOv7 on the UAVDT dataset is worse than that of YOLOv5l. Compared with YOLOv5l, the mAP0.5 is 1.2% lower, the mAP0.75 is 1.3% lower, and the mAP is 1.1% lower. However, our model outperforms YOLOv5l in terms of mAP0.5, mAP0.75, and mAP. We also listed the mAP0.5 results for each category in the table, which showed that our model improved by 2.9% over YOLOv7 in the car category, and the results for the truck and bus categories both improved by 3.4%. In addition, our model outperforms YOLOv5l by 3.3% in the truck category and is also 0.6% and 2.2% higher in the car and bus categories, respectively. At the same time, our model's performance is also superior to YOLOv3 and YOLOX.

In Table 7, we also included the results of the tiny version for comparison. Our CGMDet-tiny improved the results for the car, truck, and bus categories by 2.3%, 1.0%, and 0.9%, respectively, compared to YOLOv7-tiny. It also increased the mAP0.5 by 1.4% and improved the mAP0.75 and mAP by 2.2% and 1.8%, respectively. In addition, our CGMDet-tiny outperforms YOLOX-tiny in all metrics. However, compared to YOLOv5s, our CGMDet-tiny only outperforms by 0.3% in terms of mAP.

To illustrate the superiority of our CGMDet, we present detection results for several images in different scenarios. As shown in Figure 10(a1–a3) are the results of YOLOv7, and Figure 10(b1–b3) are the results of our proposed model. From Figure 10(a1,b1), our model performs significantly better than YOLOv7 in terms of detecting small targets. From Figure 10(a2,b2), even under low light conditions at night, our model has a significant improvement over the baseline. In addition, as shown in Figure 10(a3,b3), YOLOv7 detected the left tree as a car, and the bus as a truck, and did not detect the objects that were truncated at the bottom of the image or slightly occluded on the right side of the image. In contrast, our model correctly recognized the tree as the background, accurately identified the object categories, and accurately detected the objects at the bottom and right of the image.

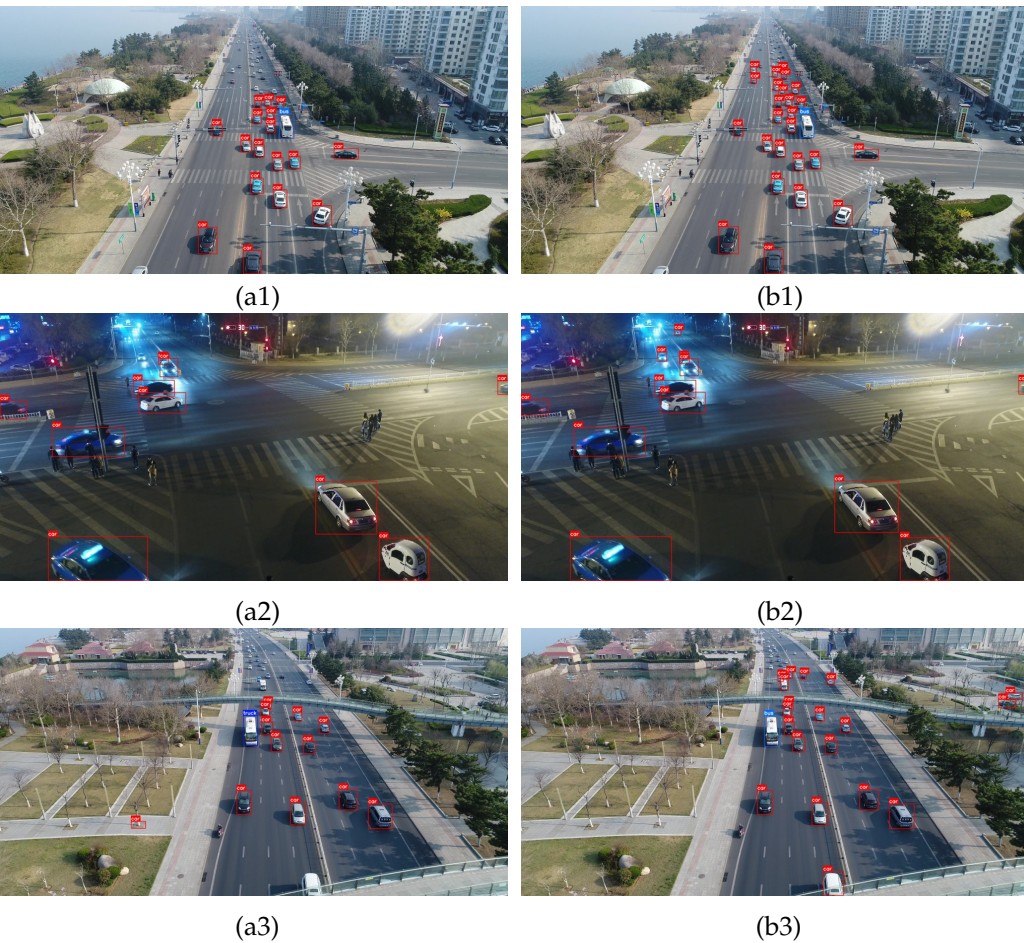

(a1) (b1)

(a2) (b2)

(a3) (b3)

**Figure 10.** Detection results of different scenes in the UAVDT dataset. (**a1**–**a3**) are the detection results of YOLOv7; and (**b1**–**b3**) are the detection results of our CGMDet.

**Table 7.** Comparison with state-of-the-art detectors on the UAVDT dataset.

| Method | Car | Truck | Bus | mAP0.5 | mAP0.75 | mAP |
|---|---|---|---|---|---|---|
| YOLOv3 [15] | 30.8 | 3.9 | 26.4 | 36.3 | 20.6 | 20.4 |
| YOLOX [45] | 39.4 | 5.7 | 25.3 | 37.9 | 26.1 | 23.5 |
| YOLOX-tiny [45] | 40.5 | 0.5 | 22.0 | 36.1 | 22.7 | 21.0 |
| YOLOv5l [17] | 80.7 | 12.7 | 45.2 | 46.2 | 30.7 | 27.7 |
| YOLOv5s [17] | 78.1 | 13.3 | 45.6 | 45.0 | 28.6 | 26.8 |
| YOLOv7 [18] | 78.4 | 12.6 | 44.0 | 45.0 | 29.4 | 26.8 |
| YOLOv7-tiny [18] | 75.5 | 6.9 | 46.7 | 43.0 | 24.3 | 25.0 |
| CGMDet (ours) | **81.3** | **16.0** | 47.4 | **48.0** | **32.6** | **29.1** |
| CGMDet-tiny (ours) | 77.8 | 7.9 | **47.6** | 44.4 | 26.5 | 26.8 |

### 4.3.3. Ablation Experiments

We used the VisDrone dataset to conduct ablation experiments for our model to verify the effectiveness of our improved methods. For fairness, all experimental settings have the same parameters and are conducted in the same environment. As shown in Table 8, we use YOLOv7 as the baseline and achieved a 49% mAP. Furthermore, the results show that each improvement can enhance the detection ability of the model to some extent.

- **CGAM**: To reflect the effectiveness of the CGAM, we replaced the ELAN [37] module in the YOLOv7 backbone with our CGAM module. Compared with YOLOv7, using CGAM increased the mAP0.5 by 0.7%. This is because our CGAM can simultaneously extract local, coordinate, and global information, making the extracted feature map richer in contextual information, and thereby improving the ability of the backbone network to extract features;
- **MF-FPN**: To demonstrate the effectiveness of MF-FPN, we replaced the neck part of YOLOv7 with the proposed MF-FPN. Compared with YOLOv7, the improved model with MF-FPN increased mAP0.5 by 1%, and the parameters of the model also decreased by 2.2M. This shows that the MF-FPN can fully integrate multi-scale features with fewer parameters. This also proves that our FFM can fully integrate features of different scales and obtain multi-scale feature maps with stronger representation ability;
- **Focal-EIOU Loss**: To reflect the effectiveness of Focal-EIOU Loss, we replaced the CIOU loss in YOLOv7 with Focal-EIOU Loss. Compared with CIOU Loss, Focal-EIOU Loss can more accurately regress the bounding box and allow high-quality anchor boxes to make more contributions during training, thereby improving detection performance. Compared with YOLOv7, the model's mAP0.5 increased by 0.5%;
- **Proposed Method**: When CGAM, MF-FPN, and Focal-EIOU loss were all incorporated into YOLOv7, our model was obtained. Compared with YOLOv7, the precision increased by 0.5%, the recall increased by 2.1%, the mAP0.5 increased by 1.9%, and the parameter size of our model was reduced by 0.7M compared to the baseline. The results show that our improvement methods are very effective, and each improvement can enhance the performance of the model;

**Table 8.** Ablation experiments.

| Method | CGAM | MF-FPN | Focal-EIOU | Precision | Recall | F1-Score | mAP0.5 | Params (M) | GFLOPs |
|---|---|---|---|---|---|---|---|---|---|
| YOLOv7 | | | | 58.3 | 49.2 | 53.4 | 49.0 | 36.5 | 103.3 |
| B1 | ✓ | | | 55.8 | 51.2 | 53.4 | 49.7 | 38.0 | 104.7 |
| B2 | | ✓ | | **61.1** | 48.2 | 53.9 | 50.0 | 34.3 | 104.0 |
| B3 | | | ✓ | 59.0 | 50.3 | 54.3 | 49.5 | 36.5 | 103.3 |
| B4 | ✓ | ✓ | ✓ | 58.8 | **51.3** | **54.8** | **50.9** | 35.8 | 105.3 |

We plotted the change process of mAP0.5 and mAP during training, as shown in Figure 11. It is evident from the figure that compared with YOLOv7, each of our improve-

ment points significantly improved the model's performance. The model we proposed by integrating all the improvement points undergoes an especially significant improvement compared to YOLOv7.

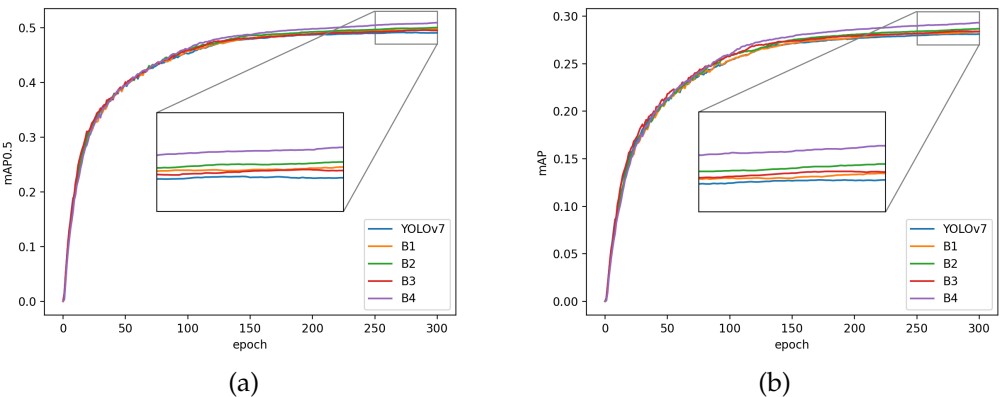

**Figure 11.** Comparison between mAP0.5 and mAP for different improvement points. (**a**) represents the result of mAP0.5; (**b**) represents the result of mAP.

In addition, we also listed the changes in the convolutional parameter sizes of each ELAN-H module in the neck part of the model. As shown in Table 1, ELAN-H_3 and ELAN-H_4 are two additional modules that we added to our model.

To better demonstrate the effectiveness of CGMDet, we used Grad-CAM [54] to visualize the model's execution results in the form of heatmaps. As shown in Figure 12, the first row of the image shows that, compared with YOLOv7, our model reduces the focus on similar objects around small targets and can more accurately detect small targets. The second row shows that our model alleviates the interference of background factors. The third row shows the heat map results generated by our model in low-light nighttime scenes, where we can observe that, even under low-light conditions, our model can accurately focus on the target while reducing the attention to the background.

We briefly tested the effects of $\lambda_1$, $\lambda_2$, and $\lambda_3$ in Formula (23) on the model's performance. As shown in Table 9, when $\lambda_1 = 0.07$, $\lambda_2 = 0.7$, and $\lambda_3 = 0.3$, mAP0.75 and mAP obtained the highest result, but mAP0.5 is 0.7% lower than the best result. When $\lambda_1 = 0.05$, $\lambda_2 = 0.6$, and $\lambda_3 = 0.3$, the mAP is also the highest, but the mAP0.5 and mAP0.75 are not very good. When $\lambda_1 = 0.05$, $\lambda_2 = 0.7$, and $\lambda_3 = 0.3$, the mAP0.5 is the highest, and the mAP0.75 and mAP are only 0.2% lower than the highest result. It can be seen from the results that, when $\lambda_1 = 0.05$, $\lambda_2 = 0.7$, and $\lambda_3 = 0.3$, the mAP0.5, mAP0.75, and mAP can achieve good balance, so we choose them as the final values.

**Table 9.** Model's performance changes of different values of $\lambda_i$.

| $\lambda_1$ | $\lambda_2$ | $\lambda_3$ | mAP0.5 | mAP0.75 | mAP |
|---|---|---|---|---|---|
| 0.03 | 0.7 | 0.3 | 49.8 | 27.7 | 28.5 |
| 0.07 | 0.7 | 0.3 | 50.2 | **29.6** | **29.5** |
| 0.05 | 0.6 | 0.3 | 50.5 | 29.3 | 29.5 |
| 0.05 | 0.8 | 0.3 | 50.6 | 29.2 | 29.4 |
| 0.05 | 0.7 | 0.2 | 49.7 | 28.9 | 29.0 |
| 0.05 | 0.7 | 0.4 | 50.6 | 28.9 | 29.3 |
| 0.05 | 0.7 | 0.3 | **50.9** | 29.4 | 29.3 |

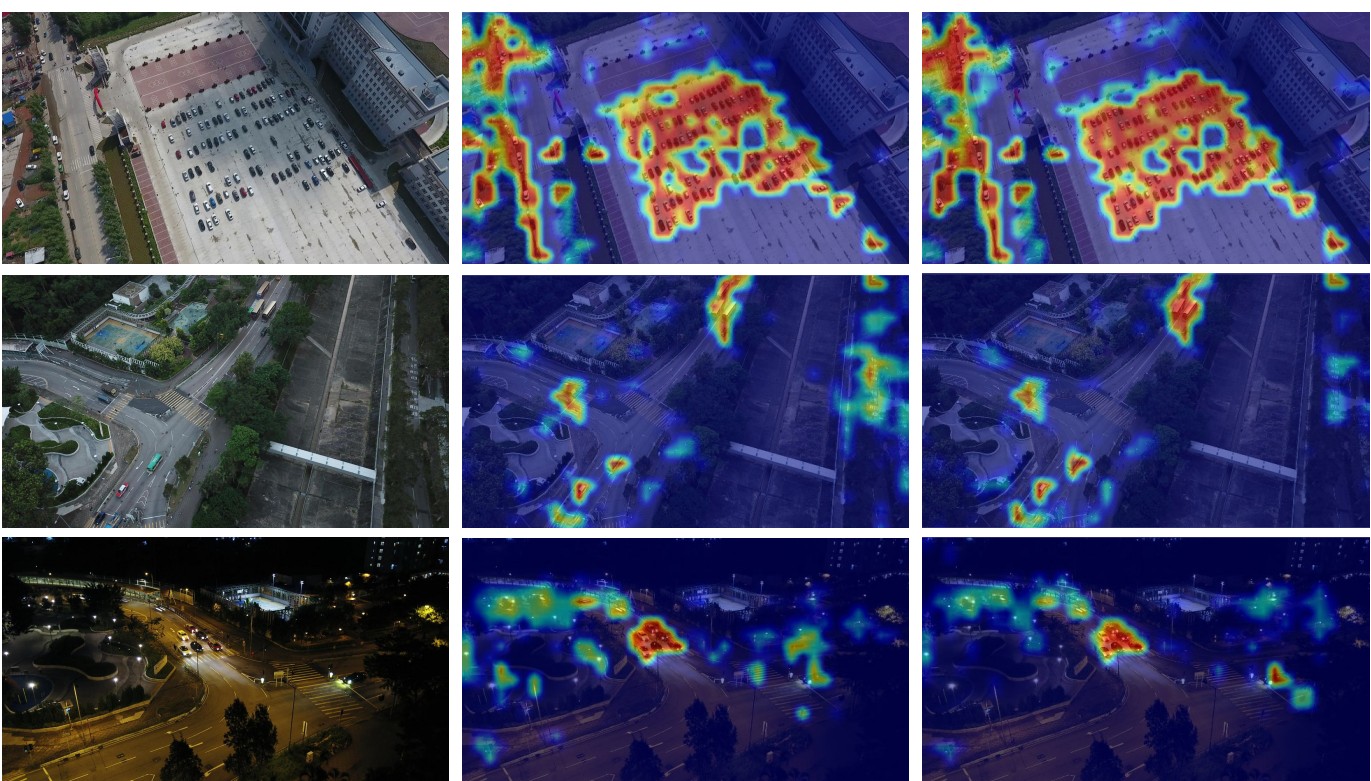

**Figure 12.** Example of heatmap visualization. The first column is the original image, the second column is the visualization result of YOLOv7, and the third column is the visualization result of CGMDet.

In addition, we explored the effects of different learning rates on the performance of our model. As shown in Table 10, the model performance gradually improves when the learning rate increases from 0.005 to 0.010. When it is higher than 0.010, the model's overall performance declines slightly. When the learning rate is 0.011, mAP reaches the highest, but mAP0.5 and mAP0.75 are lower than when the learning rate is 0.010. Similarly, when the learning rate is 0.012, mAP0.75 reaches the highest, but mAP0.5 and mAP are relatively low. Therefore, we chose 0.010 as our final learning rate.

**Table 10.** Model performance at different learning rates.

| Learning Rate | mAP0.5 | mAP0.75 | mAP |
| :---: | :---: | :---: | :---: |
| 0.005 | 48.5 | 27.9 | 28.0 |
| 0.008 | 49.4 | 28.9 | 28.9 |
| 0.009 | 50.3 | 29.0 | 29.3 |
| 0.010 | **50.9** | 29.4 | 29.3 |
| 0.011 | 50.7 | 29.1 | **29.5** |
| 0.012 | 50.4 | **29.6** | 29.4 |

### 4.3.4. Extended Experiments

To verify the generalization ability of our model, we conducted experiments on the generic dataset VOC2012 [55]. We use *VOC2012 train* for training and *VOC2012 val* for validation. The training set contains 5717 images, and the validation set contains 5823 images. The value of mAP0.5 for each category is shown in Table 11. Our model is superior to the baseline model in some categories. However, some categories are worse than the baseline model. For further analysis, we list the detection results of different scales on the VOC2012 dataset in Table 12. Our model improves by 1.8% on $AP_S$ and 0.3% on $AP_M$ but

decreases by 3.3% on $AP_L$. This shows that our model is more suitable for detecting small and medium targets.

**Table 11.** Experimental results of different categories on the VOC2012 validation set.

| Method | Horse | Person | Bottle | Dog | TV | Car | Aeroplane | Bicycle | Boat | Chair | Table |
|--------|-------|--------|--------|-----|-----|------|-----------|---------|------|-------|-------|
| Baseline | **81.1** | 83.2 | 55.3 | 80.5 | 65.6 | **78.1** | **80.4** | **79.6** | 60.4 | 56.9 | 61.5 |
| Ours | 80.1 | **83.8** | **56.9** | **81.7** | **68.0** | 77.5 | 79.2 | 77.0 | **60.5** | **57.3** | **62.9** |

| Plant | Train | Cat | Sofa | Bird | Sheep | Mbike | Bus | Cow | Head | Foot | Hand |
|-------|-------|-----|------|------|-------|-------|-----|-----|------|------|------|
| 49.7 | **84.6** | **86.9** | **66.8** | **69.9** | 74.4 | 82.6 | 82.1 | **69.0** | 13.5 | 8.9 | 10.8 |
| **50.8** | 82.5 | 86.0 | 64.9 | 68.6 | **74.5** | **83.0** | 82.1 | 64.8 | **17.7** | **15.7** | **12.2** |

**Table 12.** Experimental results of different scales on the VOC2012 validation set.

| Method | $AP_S$ | $AP_M$ | $AP_L$ |
|--------|--------|--------|--------|
| Baseline | 14.2 | 32.0 | **55.9** |
| Ours | **16.0** | **32.3** | 52.6 |

From the results of the extended experiment, our model's performance on the VOC2012 dataset is not very good, which indicates that our model is more suitable for UAV images. However, it also shows that the generalization ability of our model needs to be stronger. The result of our analysis is that after adding the P2 layer to the model for feature fusion, the feature proportion of small and medium targets increases, resulting in the model paying more attention to small and medium targets while ignoring large targets. Therefore, our model is more suitable for detecting UAV images with more small- and medium-sized targets.

## 5. Discussion

In Table 6, although the PAFPN and MF-FPN are similar in performance, our CGMDet needs to integrate more features. If we use the PAFPN, the model's number of parameters will increase. Therefore, the MF-FPN is more suitable for use in our CGMDet to avoid increasing the number of parameters in the model.

In Table 8, MF-FPN, although higher than Focal-EIOU on mAP0.5, is 0.4% lower on the F1-Score. This is because the precision of MF-FPN is increased by 2.8%, but the recall is reduced by 1.0%. However, Focal EIOU has improved both precision and recall. This shows that Focal-EIOU is better at balancing precision and recall. It can also be seen from the index F1-Score that the Focal EIOU has a better balance between precision and recall than MF-FPN, which helps improve the network's quality.

In addition to the above successes, our CGMDet has certain limitations. The performance of CGMDet on natural images is not very good, which indicates that the model's generalization ability needs to be stronger. In addition, in the UAV image, the distant target is usually densely arranged, and the target may be blocking another target. In this scenario, it is difficult for the model to determine the area and number of targets, resulting in the model generating many redundant detection boxes or detection boxes with inaccurate scopes. Therefore, the detection performance of CGMDet in this scenario needs to be improved. In addition, after the above improvements, the model complexity of CGMDet has increased, affecting the model's inference time. Therefore, the model's complexity needs to be further improved.

## 6. Conclusions

This study proposed a multi-scale object detector based on coordinate and global information aggregation for UAV aerial images. This detector can focus more on the features of the objects and better detect multi-scale objects. We designed the CGAM that integrates local, coordinate, and global information to obtain more robust features, effectively alleviating the interference of background factors such as occlusion and weak

light in the feature extraction process. The proposed FFM can integrate the features of different scales more fully by automatically learning the importance of features of different scales in fusion. Furthermore, by reusing features, the expression ability of multi-scale features is enhanced, and the detector's perception ability of different scale targets is improved. Finally, by modifying the loss function, the localization effect of the model on the target is improved. The experiments show that, compared with the baseline detector, the CGMDet improves mAP0.5 by 1.9% on the VisDrone dataset and 3.0% on the UAVDT dataset. Moreover, it can better detect multi-scale targets. In future work, we will focus on detecting dense objects and developing lightweight models while improving the generalization ability of models.

**Author Contributions:** Conceptualization, L.Z.; methodology, Z.L.; software, Z.L.; validation, L.Z., Z.L., H.Z. and Y.-E.H.; formal analysis, Y.L.; investigation, L.D.; resources, X.Z.; data curation, Y.-E.H.; writing—original draft preparation, L.Z. and Z.L.; writing—review and editing, L.Z. and Y.L.; visualization, H.Z.; supervision, H.Z.; project administration, Z.L.; funding acquisition, X.Z. All authors have read and agreed to the published version of the manuscript.

**Funding:** This work was supported by the National Basic Research Program of China (Grant no. 2019YFE0126600); the Major Project of Science and Technology of Henan Province (Grant no. 201400210300); the Key Scientific and Technological Project of Henan Province (Grant no. 212102210496); the Key Research and Promotion Projects of Henan Province (Grants no. 212102210393 and 202102110121); Kaifeng Science and Technology Development Plan (Grant no. 2002001); National Natural Science Foundation of China (Grant no. 62176087); and Shenzhen Science and Technology Innovation Commission (SZSTI), Shenzhen Virtual University Park (SZVUP) Special Fund Project (Grant no. 2021Szvup032).

**Data Availability Statement:** The data used to support the findings of this study are available from the corresponding author upon request.

**Acknowledgments:** We sincerely thank the anonymous reviewers for the critical comments and suggestions for improving the manuscript.

**Conflicts of Interest:** The authors declare no conflict of interest.

## Abbreviations

The following abbreviations are used in this manuscript:

| | |
|---|---|
| UAV | Unmanned Aerial Vehicle |
| HOG | Histogram of Oriented Gradients |
| SIFT | Scale Invariant Feature Transform |
| SSD | Single Shot MultiBox Detector |
| YOLO | You Only Look Once |
| FCOS | Fully Convolutional One-Stage Object Detection |
| CGAM | Coordinate and Global Information Module |
| MF-FPN | Multi-Feature Fusion Pyramid Network |
| SE | Squeeze and Excitation |
| ECA | Efficient Channel Attention |
| ESE | Effective Squeeze and Excitation |
| CA | Coordinate Attention |
| CBAM | Convolutional Block Attention Module |
| FPN | Feature Pyramid Network |
| PANet | Path Aggregation Network |
| MLFPN | Multi-Level Feature Pyramid Network |
| TUM | Thinned U-shape Module |
| FFM | Feature Fusion Module |
| BiFPN | Bidirectional Feature Pyramid Network |
| IOU | Intersection over Union |
| P-R curve | Precision–Recall Curve |
| GFLOPs | Giga Floating-point Operations per Second |

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
