# Peer review of "A Multi-Scale Object Detector Based on Coordinate and Global Information Aggregation for UAV Aerial Images"

_remotesensing, doi:10.3390/rs15143468_

Round 1
Reviewer 1 Report
This paper is proposed a UAV image-based object detection methods with several tricks. Honestly, there are many similar papers compared with this paper, and the performance improved by the proposed tricks cannot convince the reader that this is the SOTA work and match the standard of the remote sensing.
1. Tricks like Multi-Feature Fusion Pyramid Network have been proposed by other works for several year, it is not proper to be proposed as a novel point in this paper.
2. The other tricks are common tricks and shows no difference with other object detection methods aiming for the daily tasks. The authors should verify their tricks on daily dataset, such as COCO, ImageNet, and submit their paper to CV magazine. As for remote sensing tasks, the tricks should be more specific for the UAV based images.
3. The methods are not fully verified. The author utilizes the two public datasets to prove the effectiveness of the method, but only the general yolo based object detection methods are compared. Actually, there are many object detection methods proposed based on the UAV based dataset. For example, the PP-YOLOE-plus approaches 66.7 AP50 score on the VisDrone dataset. And the CZDet achieve the 58.3 AP50 score on the same dataset.
And there are many papers related to the Visdrone dataset detection, such as:
PP-YOLOE: An evolved version of YOLO
Slicing Aided Hyper Inference and Fine-tuning for Small Object Detection
CNN-based Density Estimation and Crowd Counting: A Survey
Object Detection in Aerial Images: A Large-Scale Benchmark and Challenges
TPH-YOLOv5: Improved YOLOv5 Based on Transformer Prediction Head for Object Detection on Drone-captured Scenarios
EdgeYOLO: An Edge-Real-Time Object Detector
A Normalized Gaussian Wasserstein Distance for Tiny Object Detection
Clustered Object Detection in Aerial Images
Delving into Robust Object Detection from Unmanned Aerial Vehicles: A Deep Nuisance Disentanglement Approach.
The author should compared at least 2 or 3 this methods to show the advance of their own method.
should be improved
Author Response
Thank you for your comments. Those comments are all valuable and very helpful for revising and improving our paper, as well as the important guiding significance to our research. We have studied the comments carefully and have made corrections which we hope meet with approval.
Please see the attachment for this paper's main changes and responses to the reviewer's comments.

Reviewer 2 Report
The paper proposes a multi-scale object detector by incorporating CGAM, MF-FPN and Focal-EIOU.
1 - CGAM needs to be compared with similar approaches
2 - MF-FPN needs to be compared with similar approaches
3 - The paper needs to be clearer on how you find the hyperparameter; you briefly mention Lines 278, 279, 302, and 303. Also, the paper does not mention the stopping criteria. I highly recommend the authors use a validation set to define the hyperparameters and show the results in the test set. I suggest the author create a validation set on UAVDT using a random subset of the training, as the dataset does not have a validation set.
4 - Equation 21. Add a multiplication symbol between IOU and L_{EIOUT}
Also in this equation. Is it an original proposal of this paper, or you just used it from another reference?
5 - Table 5. Add F1. The increment given by MF-FPN and Focal-EIOU is quite similar, showing a tradeoff between precision and recall. When we try to find the most relevant, it is hard to judge. I suggest adding an F1 column. You will see that Focal-EIOU is higher in F1. Please discuss this result.
Overall is a good paper, but CGAM and MF-FPN need to be compared with similar approaches to evaluate their contribution. Also, the hyperparameters need to be set using a validation set.
Overall is a good paper, but CGAM and MF-FPN need to be compared with similar approaches to evaluate their contribution. Also, the hyperparameters need to be set using a validation set.
Author Response
Thank you for your comments. Those comments are all valuable and very helpful for revising and improving our paper, as well as the important guiding significance to our research. We have studied the comments carefully and have made corrections which we hope meet with approval.
Please see the newest attachment for this paper's main changes and responses to the reviewer's comments.

Reviewer 3 Report
The authors design CGMDet with CGAM and MF-FPN based on yolov7 for UAV image object detection. The proposed model works well with UAV detection with the mAP improvement on multi-scale objects. In general, the manuscript has some innovations and sufficient experiments are conducted. Here are some comments.
1. The full expression and its abbreviations only coexist when first mentioned in the text.
2. In Fig2, the features from horizontal connections and bottom-up/top-down path (and skip connection) are directly fed into FFM for fusing. How are the multi-level features with different sizes fused in FFM?
3. There is no explanation for the structure of module ELAN-H. Does it follow ELAN-W on yolov7 just with the adjustment of the channel numbers? As the ELAN-H is not distinguished in Fig2, whether ELAN-H_1/2/3/4/5/6 have the same structure?
4. There is a mistake in the pseudo-code of the MF-FPN about the range of i in step2.
5. I hope you can make adequate improvements to the English language and style for the whole article.
6. In related work, some SOA object detection works should also be discussed,which are https://doi.org/10.3390/rs14030516 and https://doi.org/10.1016/j.jag.2022.102912
I hope you can make adequate improvements to the English language and style for the whole article.
Author Response

(The authors gave the same response as above.)

Reviewer 4 Report
This article addresses two challenges in UAV image target detection: (1) large differences between scales (2) boundary and features of the object are not obvious caused by blockness or weak light.
The paper mentions 5 contributions, but does not say the relationship between these 5 points and the above two challenges, especially which method solves which problem and the degree of solution. It seems that the core part is to propose a CGAM module containing two branches and build a CGMDet network based on CGAM. As for MF-FPN, it is not clearly marked where it is used in the figure. The figure also contains a large number of module abbreviations. Although the abbreviations are explained in a unified way later, it is still not clear about the specific structure and why it is designed in this way.
In summary, this paper presents an overall approach, and also provides an experimental evaluation of the approach and a comparison of general detection networks. However, there is a lack of comparison of multi-scale detection methods and multi-scale fusion modules specifically for different structures, such as MA-FPN. I think this article has a complex structure and low usability, and it does not explain why it is designed in this way, and how it improves the perception of multi-scale targets. It is recommended to add more smile comparisons, and analysis, combined with visualization methods to demonstrate the viewpoints proposed in the paper.
As for the English writing of the thesis, I can't ask more questions, and I suggest increasing the readability.
Author Response

(The authors gave the same response as above.)

Round 2
Reviewer 2 Report
The authors clarified most of my concerns.
Author Response
Dear Reviewer:
Thank you very much for your guidance and recognition of our article "A Multi-Scale Object Detector Based on Coordinate and Global Information Aggregation for UAV Aerial Images" (ID: remotessensing-2440505).
Reviewer 3 Report
The authors have improved the manucript according to the suggestions of the reviewers, so it can be accepted
Author Response

(The authors gave the same response as above.)

Reviewer 4 Report
I have seen some improvements in this revised paper, but there are still a few minor problems, please revise
1 The English font and size of the mark in Figure 5 are not uniform;
2 In Figure 2, CGAM has one input and two outputs; but in the detailed structure corresponding to Figure 3, there is only one input and one output, why?
3
The quality of English is ok.
Author Response
Dear Reviewer:
Thank you for your comments on our article "A Multi-Scale Object Detector Based on Coordinate and Global Information Aggregation for UAV Aerial Images" (ID: remotessensing-2440505). We have studied these comments carefully and made changes.
Please check the attachment for details.
